

# Difference in load predictions obtained with effective turbulence vs. a dynamic wake meandering modeling approach

Paula Doubrawa[1], Kelsey Shaler[1], and Jason Jonkman[1]

[1]National Renewable Energy Laboratory, Golden, CO 80401, USA

**Correspondence:** Paula Doubrawa (Paula.Doubrawa@nrel.gov)

**Abstract.** According to the international standard for wind turbine design, the effects of wind turbine wakes on structural loads can be considered in two ways: (1) by augmenting the ambient turbulence levels with the effective turbulence model (EFF) and then calculating the resulting loads and (2) by performing dynamic wake meandering (DWM) simulations, which compute wake effects and loads for all turbines in a farm at once. There is no definitive answer in scientific literature as to the consequences of choosing one model over the other, but the two approaches are unarguably very different. The work presented here expounds on these differences and investigates to what extent they affect the simulated structural loads. We consider an idealized 4x4 rectangular array of National Renewable Energy Laboratory 5 MW wind turbines with a spacing of 5 by 8 rotor diameters, and three wind speed scenarios at high ambient turbulence. Load simulations are performed in OpenFAST with EFF and in FAST.Farm with the DWM implementation. We compare ambient turbulence, wind farm turbulence, and loads between both approaches. When omnidirectional results are compared, EFF wind farm turbulence intensity is consistently higher by 0.2% (above rated wind speed) to 2.7% (below rated wind speed). However, for certain wind directions, the EFF turbulence can be lower than FAST.Farm by almost 9%. Wind speeds within the farm were found to differ by up to 3 m s$^{-1}$ due to the lack of wake deficits in the EFF approach, leading to longer tails toward low values in the FAST.Farm mean load distributions. Consistent with the turbulence results, the median EFF load standard deviations are also consistently higher, by a maximum of 20% and 17% for blade-root out-of-plane and tower base fore-aft moments, respectively.

## 1 Introduction

Modeling approaches that have become standard industry practice tend to be relatively simple to implement and inexpensive to run. Over time, as simulation tools improve and computational costs decrease, modeling approaches that are incrementally more complex and expensive become increasingly accessible. When that happens, we have the opportunity to revisit standard procedures and evaluate whether the wind industry as a whole might benefit from updated practices that incorporate higher-fidelity models into the design process.

The International Electrotechnical Commission (IEC) standard (International Electrotechnical Commission, 2019) recommends two methods to capture wake effects in wind turbine design and site suitability analyses: the effective turbulence model (EFF; Frandsen (2007); Frandsen and Thøgersen (1999)) and the dynamic wake meandering (DWM) model (Larsen et al., 2008). The EFF is a much simpler approach that encapsulates wake effects into increased (i.e., "effective") turbulence levels





relative to the freestream. The DWM model is more complex, requiring wind farm simulations that resolve the wake dynamics of individual turbines with simplified physical and empirical parameterizations. This difference in complexity means that running DWM simulations requires more expertise, time, and computational resources. However, the DWM model can still be run on a single computer node, and the effort levels are fairly low when compared to high-fidelity methods such as large-eddy simulations.

If we agree that the two modeling approaches—EFF and DWM—are computationally tractable, then the next obvious question is whether the added investment of running DWM simulations is worth the effort and cost. The work presented here helps answer this question. To do so, we quantify the differences between both methods throughout each step in the simulation chain: from inflow generation to loads predictions. We use the effective turbulence definition found in the IEC standard (International Electrotechnical Commission, 2019) and the DWM model as implemented in FAST.Farm (Jonkman and Shaler, 2021). In addition to quantifying differences in turbulence levels, load means, and fluctuations, we quantify the computational cost of both methods.

Even if we assume that the higher-fidelity DWM model is more accurate, choosing it over the EFF will depend on the magnitude of their differences and the acceptable margin of error for the application in question. There is no obvious consensus in scientific literature as to the magnitude and sign of EFF errors relative to measurements or higher-fidelity models. The first studies comparing the EFF to other reference data (Thomsen et al., 2007; Sørensen et al., 2008; Schmidt et al., 2011a, b) identified overestimations and underestimations of loads depending on the incoming wind speed, the interturbine spacing, and the turbine component being considered. Later studies seemed to agree on an overestimation of wind farm turbulence levels (Argyle et al., 2018; Reinwardt et al., 2018) and fatigue loads (Reinwardt et al., 2018; Slot et al., 2018, 2019). In terms of load components, errors seem consistently larger for the tower than for the blades (Thomsen et al., 2007; Schmidt et al., 2011b; Reinwardt et al., 2018; Slot et al., 2019). Slot et al. (2019) was the only study to include main shaft torque, and they found those errors to be larger than those of the tower. With regard to fatigue error magnitudes, studies report a large range: $\sim 3\%$ (Figure 3 in Slot et al. (2019)) to $\sim 37\%$ (Figure 11 in Reinwardt et al. (2018)) for the tower base, with some studies finding values in between (i.e., 20% in Thomsen et al. (2007) and Schmidt et al. (2011b)).

Three existing studies have compared load predictions between the EFF and DWM implementations. Each of these studies used a different aeroelastic simulation tool: HAWC in Thomsen et al. (2007), Bladed in Schmidt et al. (2011a, b), and FLEX5 in Reinwardt et al. (2018). No details are provided in these studies as to the inflow generation methods used. For a fixed turbulence level, Thomsen et al. (2007) found the EFF to overestimate tower and blade loads at higher winds ($20 \text{ m s}^{-1}$) and underestimate them at lower winds ($10 \text{ m s}^{-1}$). The largest differences were on the order of 27% for fatigue loads and 65% for extreme loads. When considering a realistic wind climate and lifetime calculations, Schmidt et al. (2011b) did not find any underestimations on the part of the EFF. Rather, EFF fatigue loads were higher by $\sim 10\%$ for the blades and $\sim 20\%$ for the tower. Finally, Reinwardt et al. (2018) found overpredictions higher than $35\%$ for partially waked conditions when considering closely spaced ($< 4$ rotor diameters) wind turbines.

The work that we present here was motivated by the small number of published studies on this topic and the lack of consistency among them. We compare the EFF to the DWM implementation within FAST.Farm, which was previously validated



against large-eddy simulations and measurements (Kretschmer et al., 2021; Shaler and Jonkman, 2021; Shaler et al., 2020). We analyze the results after each step in the simulation chain, which allows us to identify the sources of differences between the two approaches. The simulation and analysis methods are described in Sect. 2. The results are presented in terms of freestream turbulence, wake turbulence, and loads in Sect. 3. A final discussion is given in Sect. 4. We hope that the results we present will allow wind farm modeling experts to be informed and intentional when choosing simulation tools and making design decisions.

## 2 Methods

This section describes the two simulation approaches used to estimate wind turbine loads (Sect. 2.1), the wind farm layout considered (Sect. 2.2), and the inflow scenarios defined (Sect. 2.3).

### 2.1 Simulation approaches

The structural loading on each wind turbine is simulated with two different methods: the EFF and a DWM approach as implemented in FAST.Farm (Jonkman and Shaler, 2021). The EFF is the more simplified and inexpensive approach and has, for these reasons, become the industry standard for wind plant fatigue load assessment. As the name implies, the DWM approach simulates the dynamic interactions between inflow, turbines, and wake. Despite its higher complexity and computational cost, it is still inexpensive enough to be appropriate for design calculations. Broadly speaking, the main difference between these methods is how they treat the effects of wind turbine wakes on the structural loads. More specific details on how the two methods differ are discussed next.

#### 2.1.1 Effective turbulence

The EFF implemented here is based on the equations provided in the IEC design standard for wind turbines (International Electrotechnical Commission, 2019). The effective turbulence $I_{eff}$,

$$I_{eff} = \frac{\sigma_{eff}}{V_{hub}} \tag{1}$$

is obtained from the hub-height wind speed $V_{hub}$ and effective standard deviation $\sigma_{eff}$,

$$\sigma_{eff} = \left[ (1 - N * 0.06)\sigma_c^m + 0.06 \sum_{i=1}^{N} \sigma_T^m \right]^{m^{-1}} \tag{2}$$

where $N$ is the number of neighboring wind turbines, 0.06 is the probability of wake conditions (Sørensen et al., 2008), $\sigma_c$ is the characteristic wind speed standard deviation, $m$ is the Wöhler exponent, and $\sigma_T$ is given by

$$\sigma_T = \left[ \frac{V_{hub}^2}{1.5 + \frac{0.8d}{\sqrt{C_T}}} + \sigma_c^2 \right]^{1/2} \tag{3}$$



where $d$ is the distance between the target turbine (i.e., the one for which $\sigma_{eff}$ is being computed) and neighboring turbine $i$ and $C_T$ is the wind turbine thrust coefficient. Here, three wind and thrust values are used based on the wind scenarios defined (Sect. 2.3.1). For each scenario, we also consider three Wöhler exponents: $m = 4$ for the tower (welded steel), $m = 6$ for the nacelle components (cast steel), and $m = 10$ for the blades (fiber composites). The output of the model is the effective wind speed standard deviation ($\sigma_{eff}$) at each wind turbine, combining the wake effects of its neighbors for all wind directions relative to the wind farm. A uniform wind direction distribution is considered for both simulation approaches. After obtaining the $\sigma_{eff}$ values, turbulent inflow and wind turbine simulations are then performed independently for each wind turbine in the wind farm. We use TurbSim (Jonkman, 2009) to generate synthetic turbulent inflow based on a Kaimal spectrum with exponential coherence and OpenFAST (NREL, 2023) to simulate the wind turbine dynamic response, including aero-servo-elastics.

### 2.1.2 FAST.Farm

FAST.Farm is a wind farm simulation tool that expands on DWM principles to model wind turbine performance and structural loading within wind farms. Within FAST.Farm, the entire farm is simulated at once with one OpenFAST instance for each wind turbine. Each OpenFAST instance in the FAST.Farm simulations is set up identically to the OpenFAST simulations performed in the EFF approach. In addition to these OpenFAST instances, FAST.Farm computes wake deficit evolution, advection, deflection, meandering, and merging. A snapshot of flow fields simulated with FAST.Farm for the work performed here is shown in Fig. 1. More information on its capabilities and models can be found in Jonkman and Shaler (2021). Note that wake-added turbulence is a forthcoming capability of FAST.Farm that was not available in the model version used here. That said, the ambient turbulence intensities simulated in the wind scenarios are high enough that the absence of wake-added turbulence would not likely impact the conclusions of this study (Shaler and Jonkman, 2021).

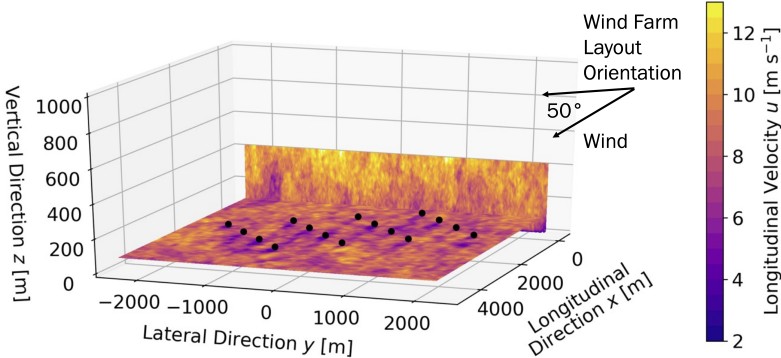

**Figure 1.** Snapshot (300 seconds into a 600-second simulation) of flow fields simulated by FAST.Farm for the below-rated wind speed scenario at a 50° orientation between the farm layout and the incoming wind. Black dots mark the turbine locations.

In FAST.Farm, we explicitly consider the effects of varying wind direction on wake dynamics and loads. This is done by performing separate FAST.Farm simulations for every possible incoming flow orientation relative to the wind farm layout. To avoid simulating separate inflows for each of these orientations, we choose to keep the same inflow direction (toward $+x$)





and rotate the wind farm within it. We consider 19 wind farm rotation angles: every $5°$ from $0°$ to $90°$. The results from these
simulations are then broadcast to a full wind rose by leveraging the symmetry of the rectangular wind farm layout. More details
on this broadcasting process are provided in Sect. 2.2. Specific details of the simulation setup are given in Appendix A.

### 2.1.3 Differences between approaches

For both methods, the undisturbed inflow and turbine models are identical. The only difference between them is how the inflow
to specific wind turbines is calculated. This difference affects several aspects of the simulation, as summarized in Table 1. The
differences in computational cost are also given in Table 1, with FAST.Farm requiring $\sim 14\times$ more time for inflow generation
(primarily due to the farmwide inflow simulations) and $\sim 7\times$ more time for loads simulations.

## 2.2 Wind farm layout

We consider an idealized 4x4 rectangular wind farm composed of 16 National Renewable Energy Laboratory 5 MW wind
turbines (hub height $z_{hub}$ = 90 m and rotor diameter $D$ = 126 m) with fixed land-based foundations in flat terrain. The inter-
turbine spacing is $8D$ along $x$ and $5D$ along $y$ (Fig. 2a). As mentioned in Sect. 2.1.2, wind direction effects are added to the
FAST.Farm results by performing 19 sets of simulations where the wind farm orientation with respect to the incoming wind
changes in increments of $5°$. The wind turbine rotors always remain aligned with respect to the incoming mean flow direction.
Four examples (for wind farm rotation angles of $0°$, $25°$, $50°$, and $90°$) are shown in Fig. 2.

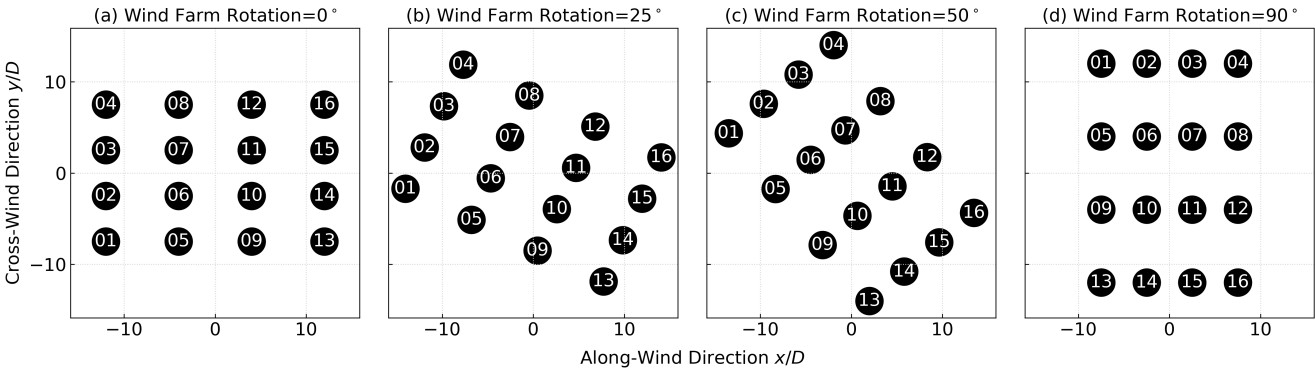

**Figure 2.** (a) Wind farm layout and (b–d) three example rotation angles out of the 19 simulated in FAST.Farm to account for wind direction
effects on the results. Regardless of wind farm rotation, the incoming flow is always from $-x$ to $+x$. The wind turbine markers are not scaled
with the rotor diameter. The wind turbine numbering shown is valid for the 19 simulation setups.

In order to account for wind direction effects across the entire $360°$ circle, the results from these 19 simulations are broadcast
from one to four quadrants. This is made possible by the rectangular layout of the wind farm. For example, a wind farm
orientation of $0°$ is equivalent to one of $180°$. Likewise, an orientation of $15°$ is equivalent to $-15°$ ($345°$), $180° + 15°$ ($195°$),
and $180° - 15°$ ($165°$). The $0°$ and $90°$ simulation results get duplicated (to $180°$ and $270°$) while those between $5°$ and $85°$ get





**Table 1.** Differences between the two modeling approaches considered.

| | Effective Turbulence Model (EFF) | FAST.Farm |
|---|---|---|
| Turbulent inflow | Stochastic turbulence is generated with TurbSim for each wind turbine using undisturbed inflow mean wind speed and effective (i.e., augmented for wake effects) turbulence levels. | Stochastic turbulence is generated with TurbSim for the wind farm domain and each wind turbine using undisturbed inflow wind speed and turbulence levels. |
| Impact of wakes on the inflow to each wind turbine | Increased turbulence is the only wake effect that is considered in the loads calculation at each turbine. | Any changes to the flow brought on by the presence of wakes (wake deficits, advection, deflection, meandering, and merging are included in the version of FAST.Farm used here) are able to impact the loads. |
| Coupling between wakes and loads calculations | The bulk, parameterized effect of wakes is described solely by a higher turbulence level, which is used to drive load simulations without any explicit treatment of wake dynamics. | Dynamic wakes are coupled to load calculations online (i.e., both are computed at the same time in an unsteady wind farm simulation). |
| Impact of neighboring wakes on resulting loads | Only wakes from adjacent wind turbines can affect the turbulence levels at a target wind turbine. | Wakes from any upwind wind turbines might affect the local flow field (mean and turbulence fields) at a target wind turbine depending on flow dynamics. |
| Impact of wind direction on wakes | Not explicitly calculated. A single effective turbulence level is obtained for each wind turbine, representing the combined effect from all wind directions. | Explicitly calculated by performing a separate wind farm simulation for each possible combination of wind direction, represented by wind farm orientations. |
| Impact of wind direction on loads | The directional integration of turbulence prior to the load simulations means that only an omnidirectional load response can be obtained with this method. | A different load response is obtained for each wind direction, represented by wind farm orientations. |
| Computational cost for inflow simulations | 2 node-hours (includes 3 inflows, 10 seeds, 16 turbines, 3 Wöhler exponents). | 28 node-hours (includes 3 inflows, 10 seeds, 16 turbines, 19 orientations and both low-resolution (farmwide) and high-resolution (turbine-specific) domains). |
| Computational cost for loads simulations | 27 node-hours (includes 3 inflows, 10 seeds, 16 turbines, 3 Wöhler exponents). | 185 node-hours (includes 3 inflows, 10 seeds, 19 orientations). |

quadruplicated, leading to a total of 72 simulation results out of the 19 originals. When broadcasting the $0°$–$90°$ quadrant to the three other quadrants, the wind turbine numbering also needs to be appropriately reconsidered. For example, Turbine 1 (T1) in the original layout (Fig. 2a) does not get waked for any of the 19 simulations performed. However, that same wind turbine will be waked if the wind direction remains constant (toward $+x$) and the wind farm is rotated past $90°$. To appropriately account for that, the wind farm numbering is mirrored across the $x$-axis, $y$-axis, or both, depending on the quadrant being considered.

130





Figure 3 specifies the mapping of wind turbine numbering from the original layout (Fig. 2a) in quadrant 1 ($0°–90°$) to farm rotations in the other three quadrants.

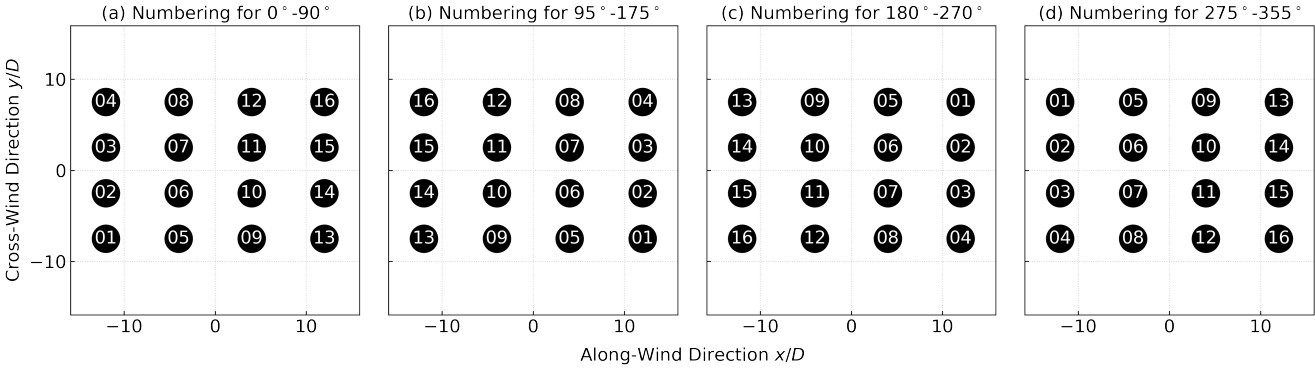

**Figure 3.** Wind turbine mapping used to broadcast the 19 simulation results spanning $0°–90°$ to the other three quadrants.

### 2.3 Inflow

#### 2.3.1 Inflow scenarios

We consider three different inflow scenarios: below ($V_{hub} = 8 \text{ m s}^{-1}$, $I_{hub} = 15.4\%$, $C_t = 0.79$), near ($V_{hub} = 12 \text{ m s}^{-1}$, $I_{hub} = 13.1\%$, $C_t = 0.54$), and above ($V_{hub} = 18 \text{ m s}^{-1}$, $I_{hub} = 12.5\%$, $C_t = 0.14$) rated where $V_{hub}$ and $I_{hub}$ are undisturbed (i.e., freestream) wind speed and turbulence intensity, respectively, at the wind turbine hub height. The wind direction is assumed to be uniformly distributed. The vertical shear follows a power law with an exponent of 0.2. This section describes how the $I_{hub}$ values used in each scenario were determined to ensure maximum consistency between the two simulation approaches. All values relevant to the inflow turbulence calculation are provided in Table 2. The significance of each value will become clear throughout this section as the inflow generation methods are discussed and contrasted.

The $I_{hub}$ values that correspond to each inflow scenario were constrained in two ways. First, they were chosen so that any augmented turbulence levels (found deep in the farm) would still remain within the wind turbine design envelope while assuming a normal turbulence model and a Class A site (International Electrotechnical Commission, 2019). Second, the desired values could not be obtained exactly due to simulation tool (TurbSim) constraints. Next, we explain all steps that were taken to determine the desired turbulence levels ($\sigma_{c,des}$, $I_{hub,des}$, $I_{mid,des}$) given in Table 2.

1. Define maximum bound on the effective standard deviation ($\sigma_{eff,max}$): For each $V_{hub}$ value selected, the design turbulence standard deviation $\sigma_1(V_{hub})$ is computed for turbulence Class A as $\sigma_1 = 0.16(0.75V_{hub} + 5.6 \text{ ms}^{-1})$ $and$ $taken$ $as$ $\sigma_{eff,max}$.

2. Define maximum bound on the characteristic standard deviation ($\sigma_{c,max}$): A range of representative ambient turbulence standard deviation $\sigma_c < \sigma_1$ is used to compute $\sigma_{eff}$ for a wind turbine in the center of the defined farm, where the EFF





**Table 2.** Starting, intermediate, and final values of the turbulence calculation used when defining the three inflow scenarios considered in this study. Symbols in the table represent inflow wind speed at hub height ($V_{hub}$) and at the vertical center of the TurbSim grid ($V_{mid}$); maximum acceptable effective standard deviation ($\sigma_{eff,max}$); maximum acceptable characteristic standard deviation ($\sigma_{c,max}$); desired characteristic standard deviation ($\sigma_{c,des}$); obtained characteristic standard deviation ($\sigma_{c,obt}$); desired turbulence intensity at hub height ($I_{hub,des}$) and at the vertical center of the TurbSim grid ($I_{mid,des}$); obtained turbulence intensity at hub height ($I_{hub,obt}$) and at the vertical center of the TurbSim grid ($I_{mid,obt}$).

| $V_{hub}$ | $V_{mid}$ | $\sigma_{eff,max}$ | $\sigma_{c,max}$ | $\sigma_{c,des}$ | $\sigma_{c,obt}$ | $I_{hub,des}$ | $I_{mid,des}$ | $I_{hub,obt}$ | $I_{mid,obt}$ |
| (m s$^{-1}$) | (m s$^{-1}$) | (m s$^{-1}$) | (m s$^{-1}$) | (m s$^{-1}$) | (m s$^{-1}$) | (%) | (%) | (%) | (%) |
|---|---|---|---|---|---|---|---|---|---|
| 8.0 | 9.48 | 1.86 | 1.65 | 1.32 | 1.24 | 16.5 | 13.9 | 15.4 | 13.0 |
| 12.0 | 14.22 | 2.34 | 2.06 | 1.65 | 1.57 | 13.7 | 11.6 | 13.1 | 11.0 |
| 18.0 | 21.32 | 3.06 | 2.94 | 2.35 | 2.24 | 13.1 | 11.0 | 12.5 | 10.6 |

model will yield the highest effective turbulence. The $\sigma_c$ value that yields $\sigma_{eff} = \sigma_{eff,max}$ is taken as $\sigma_{c,max}$, as shown in Fig. 4.

3. Define desired characteristic standard deviation ($\sigma_{c,des}$): To be conservative, instead of using $\sigma_{c,max}$ to drive the simulations, we choose to use 80% of this value. This choice maintains a high turbulence level that is relevant for fatigue loading while still staying below the design constraints. Therefore, $\sigma_{c,des} = 0.8\sigma_{c,max}$.

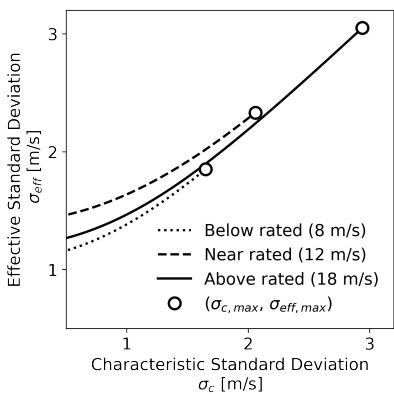

**Figure 4.** Sensitivity of effective standard deviation ($\sigma_{eff}$) to characteristic ambient standard deviation ($\sigma_c$) for each wind speed being considered and a Wöhler exponent of $m = 10$. The circles mark the maximum acceptable characteristic value $\sigma_{c,eff}$ to keep the effective value $\sigma_{eff,max}$ within the design bounds.

Once the desired turbulence levels are known, the first inflow generation steps can be taken. A summary of all simulation steps, showing the order in which they are conducted, is given schematically in Fig. 5. In FAST.Farm, the turbulent inflow
(ambient wind) is generated once for the entire farm so that all wind turbines can be simulated at the same time. TurbSim





is run separately for the farmwide and wind turbine-specific domains, but FAST.Farm uses all inflows together during the computations. The farmwide inflow runs are performed first and subsequently used to drive the turbine-specific inflow runs to ensure so that they are consistent (the turbine-specific inflows are driven by the farmwide inflows via the constrained turbulence functionality of TurbSim). Given this procedure, the first step in inflow generation (Step FF.1 in Fig. 5) is to use the computed

$\sigma_{c,des}$ values to drive the farmwide TurbSim simulations for FAST.Farm. The farm domain was run 30 times: 10 random seeds for 3 different inflow scenarios. Ten seeds were used to obtain more statistical convergence than would be obtained by the IEC recommendation of six seeds. More details on the TurbSim setup for these simulations is given in Appendix B.

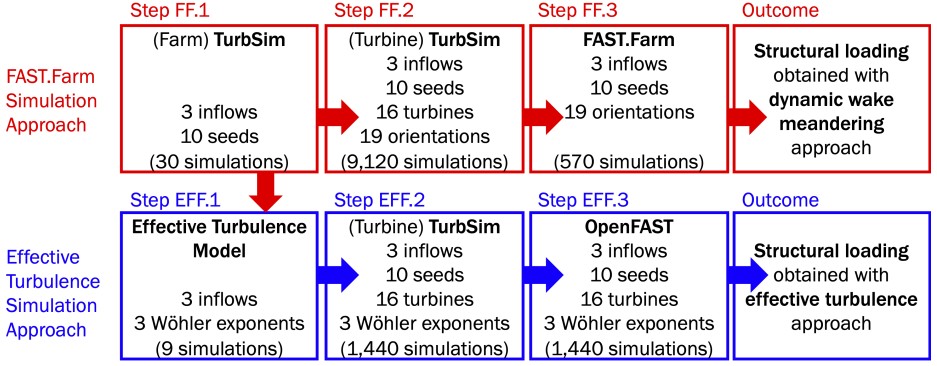

**Figure 5.** Schematic of methodology used to estimate structural loading on each wind turbine with the two different modeling methods.

The turbulence levels obtained from TurbSim in Step FF.1 are then quantified and used to drive the effective turbulence model (Step EFF.1 in Fig. 5). These obtained $\sigma_{c,obt}$ values (i.e., produced by TurbSim) might differ from the desired $\sigma_{c,des}$

values (i.e., requested from TurbSim) for two reasons. First, the TurbSim code cannot enforce a specific turbulence level at any height that is lower than half of the grid height. Therefore, we cannot enforce a specific value at the hub height of 90 m given that our grid height is set to 420 m; thus, the reference height at which turbulence levels are requested is the vertical center of the grid, $z_{mid} = 210$ m. The farmwide domain must be wide and tall to accommodate any lateral and vertical wake deflection and meandering in addition to the farm rotations performed to account for wind direction effects. Second, the turbulence levels

produced by TurbSim might differ by a few percent depending on the random seed from the synthetic generation approach that is based on a finite number of frequency components. An example of this mismatch is shown in Fig. 6 where the requested value $\sigma_{c,des}$ is shown along with the lateral distribution and lateral average of $\sigma_u$.

Instead of picking a specific lateral location in the farmwide domain to evaluate the obtained standard deviation $\sigma_{c,obt}$, we average all values laterally at hub height. This process yields 10 standard deviation values (one per turbulent seed) per inflow

scenario, which are given in Fig. 7 at $z_{mid}$ and $z_{hub}$. The seed-averaged standard deviation (marked by the horizontal dashed line in Fig. 7 and given numerically in Table 2) is then taken as the $\sigma_{c,obt}$ that is used to drive the effective turbulence model in Step EFF.1. Results from this first effective turbulence simulation step are shown in Fig. 8. By obtaining $\sigma_{c,obt}$ from the

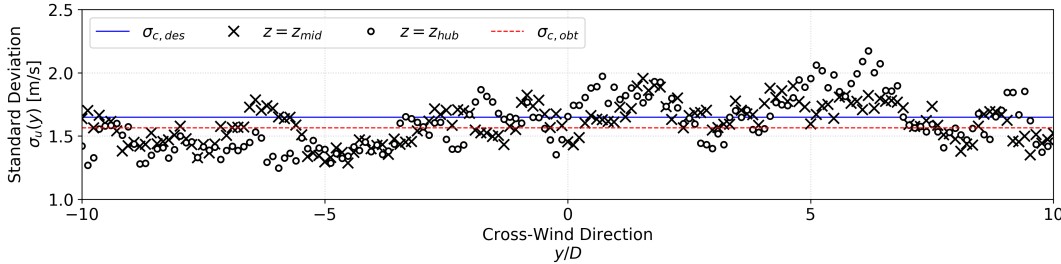

**Figure 6.** Standard deviation of longitudinal wind velocity for farm-domain TurbSim simulation for one of the random seeds and the near-rated inflow scenario. Values are given at the vertical center of the TurbSim grid ($z_{mid} = 210$ m) and at the wind turbine hub height ($z_{hub} = 90$ m) as a function of the cross-wind direction (which is limited to $\pm 10D$ for clarity). The dashed horizontal lines mark the desired ($\sigma_{c,des}$) and obtained ($\sigma_{c,obt}$) characteristic standard deviation (averaged along the cross-wind direction) for this specific inflow scenario and seed.

TurbSim simulations, we ensure that, to the extent possible, FAST.Farm and the EFF are driven with the same inflow wind speed and turbulence levels.

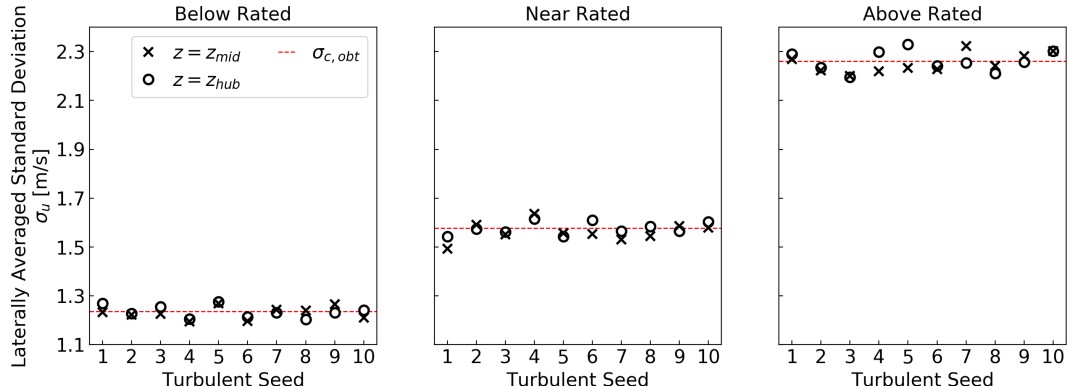

**Figure 7.** Standard deviation of longitudinal wind velocity for farmwide domain TurbSim simulations averaged laterally and given at the vertical center of the TurbSim grid ($z_{mid} = 210$ m) and at the wind turbine hub height ($z_{hub} = 90$ m) for each of the 10 random seeds. The horizontal lines give the averaged value across all seeds ($\sigma_{c,obt}$).

The flow fields generated with Step FF.1 (farm-domain TurbSim runs) are also used to drive Step FF.2 (turbine-specific TurbSim runs). This step needs to include all 16 wind turbines in the farm and all 72 park orientations, which are proxies for varying wind directions. The wind direction did not need to be considered for the farmwide domain TurbSim inflow files because the domain is large enough to accommodate any park rotation. However, the turbine-specific TurbSim domains are small and therefore need to be generated separately for each rotation angle because of the changing wind turbine locations within
the farmwide domain. A total of 9,120 high-resolution TurbSim simulations are performed in Step FF.2: one for each wind



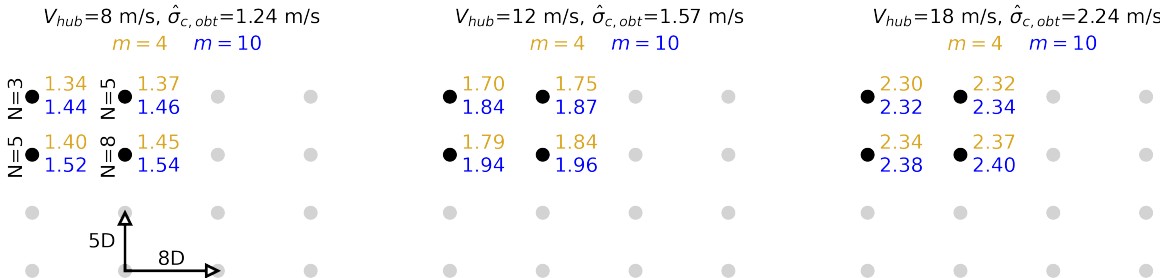

**Figure 8.** Effective turbulence model (EFF) results in terms of effective standard deviation for the three inflow scenarios when using the lowest ($m = 4$, yellow) and highest ($m = 10$, blue) Wöhler exponents considered. For clarity, results are only shown for a corner of the wind farm. The other corners are symmetrically identical. Arrows show spacing along $x$ ($8D$) and $y$ ($5D$). The $N$ value provided refers to the number of neighboring wind turbines, which is used in the calculations.

turbine (16), inflow scenario and seed (3*10), and park orientation (19). To ensure consistency between the farmwide inflow and the turbine-specific inflows, we use the constrained turbulence functionality (time series input option) of TurbSim when generating the turbine-specific inflow files. In practical terms, this means that for each inflow, seed, and park configuration, the low-resolution TurbSim data are sampled at the point closest to the turbine hub. Vertically, that ends up being exactly at hub height (90 m) for all cases because all farmwide domain TurbSim grids have a point at the turbine hub height. Laterally, that might end up being a few meters off the turbine hub location to either side due to the relatively coarse resolution of the farmwide domain TurbSim grid. The axial location of the wind turbine ($x_{WT}$) is converted to a position in the time domain $t_{WT} = x_{WT}/V_{mid}$, where $V_{mid}$ is the advection speed of the flow field within FAST.Farm.

Finally, TurbSim is also run for the effective turbulence approach (Step EFF.2 in Fig. 5). In total, 1,440 TurbSim runs were performed to drive the effective turbulence load simulations: one for each wind turbine (16), inflow scenario and seed (3*10), and Wöhler exponent (3). The grid size and resolution, time step, and duration are identical to the turbine-specific domain simulations that were run to generate inflow for FAST.Farm. Other run-time options follow the design standard recommendations for the turbulence model and coherence parameters. The turbulence levels requested from each simulation (via the TurbSim parameter `IECTurbc`) are determined from the EFF results by normalizing the effective standard deviation (for a given wind turbine, inflow scenario, and Wöhler exponent) by the freestream wind speed, $\sigma_{eff}/V_{hub}$. For example, the top left wind turbine (T4 in Fig. 2) in the below-rated scenario is driven to `IECTurbc` $= 1.33/8 = 16.6\%$ for $m = 4$ and to `IECTurbc` $= 1.43/8 = 17.9\%$ for $m = 10$ (refer back to Fig. 8 to understand these numbers). Note that here, the run-time option `ScaleIEC=1` was activated, which means that the obtained turbulence level at the hub exactly matches the requested turbulence level. The same was not possible when generating turbine-specific inflow for FAST.Farm, where the only constraint was a matching time series between the farm-domain and turbine-domain runs.



# 3 Results

## 3.1 Turbulence levels

To facilitate the interpretation of the fatigue load results, it is first important to understand the differences in turbulence levels between the two simulation methods. Here, we describe the source of these differences and quantify them in terms of the

215 undisturbed (i.e., before consideration of wake effects) and disturbed (i.e., with wakes) inflow.

### 3.1.1 Turbulence before wakes

In the effective turbulence approach, the freestream turbulence at the hub of each turbine is characterized by a single $\sigma_{c,obt}$ value per inflow scenario. In Fig. 9, this value is normalized by $V_{hub}$ and given in terms of turbulence intensity. Conversely, the freestream turbulence experienced at the hub by each turbine in the FAST.Farm approach varies substantially across space

and from simulation to simulation. First, recall that the freestream $\sigma_{c,obt}$ from the farm-domain simulations is not a single value per inflow scenario as it is in the effective turbulence approach. Rather, it is a function of the random seed and spatial location (as exemplified in Figs. 6 and 7). These various values might still change slightly when the turbine-specific domain TurbSim simulations are performed. These changes happen only for wind turbine locations that do not coincide exactly with an existing point in the coarser, farmwide domain grid, thereby requiring TurbSim to generate new turbulence at the requested

points. The distributions of these freestream turbulence hub-height values are shown in Fig. 9 for the three inflow scenarios. The distribution means and interquartile range are also shown. Note that the turbulence variability assessment presented here refers only to the hub point. Both modeling approaches still exhibit turbulence variability across the rotor area.

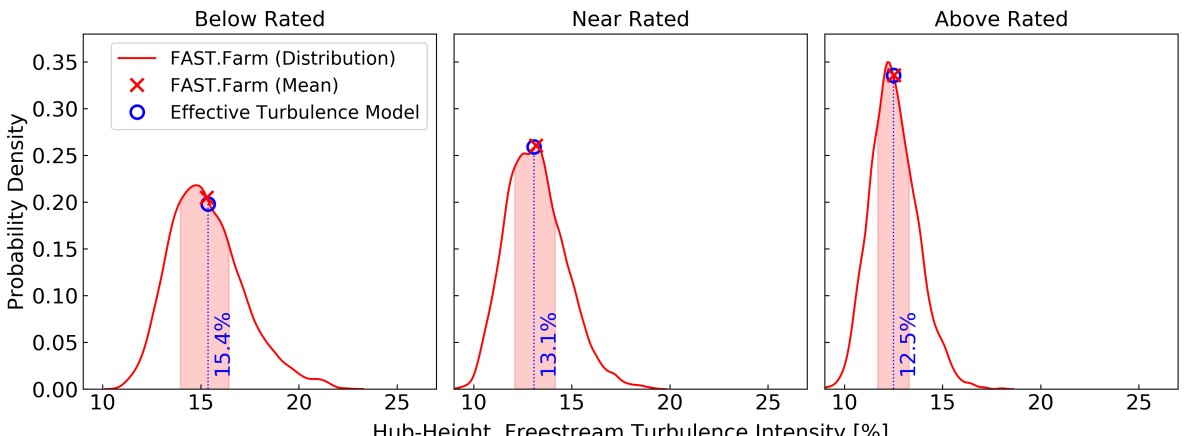

**Figure 9.** Probability density of hub-height, freestream turbulence intensity values at each wind turbine in the FAST.Farm approach (red line). Each distribution includes 11,520 data points: 10 random seeds, 72 wind farm orientations, and 16 wind turbines. The shaded red area under the curve marks the 25th to 75th percentiles of the distribution. The red x marks the mean. The hub-height, freestream turbulence intensity in the effective turbulence model (EFF) is the same for all seeds and wind turbines, as shown by the blue circle and given in text.





The values plotted in Fig. 9 highlight the variability in the freestream turbulence driving FAST.Farm, which is not present in the EFF. The probability density curves include spatial variability (across the 16 wind turbines), wind direction variability (across the 72 wind farm/wind direction orientations) and simply turbulence variability (across the 10 random seeds). Variability aside, the mean values from FAST.Farm and the constant value in the EFF match very well for the three inflow scenarios, confirming that the methodology used to generate freestream turbulence was successful in ensuring consistency to the extent possible between the two approaches.

### 3.1.2 Turbulence with wakes

As previously discussed (Table 1), wake effects on turbulence are considered differently in the two simulation strategies. In FAST.Farm, they are computed within the wind farm simulation because the freestream inflow drives the wind turbines to operate (Fig. 5, Step FF.3). In the effective turbulence approach, wake effects are accounted for via the EFF (Fig. 5, Step EFF.1). The effective standard deviation results obtained were already shown in Fig. 8 for the lowest and highest Wöhler exponents considered. These $\sigma_{eff}$ values were then used to produce the stochastic turbulence flow fields (Fig. 5, Step EFF.2), which drove the wind turbine simulations (Fig. 5, Step EFF.3). The comparison presented here considers the turbulence levels at the outcome (Fig. 5) of the two modeling approaches. More specifically, the turbulence intensity shown is computed as the coefficient of variation for the time series of hub-height longitudinal velocity at each wind turbine as output by OpenFAST (in the effective turbulence approach) and FAST.Farm. The results are shown in Fig. 10.

The directional variability in the FAST.Farm results is clear and consistent with expectations. Front-row wind turbines see turbulence levels close to the average freestream turbulence intensity (given by $I_{hub,obt}$ in Table 2 and shown as a dashed black line in Fig. 10). Second- to fourth-row turbines see turbulence levels higher than freestream due to the presence of wakes. Turbulence is highest for waked turbines when the flow is aligned with a row in the farm. This effect is most pronounced for the below- and near-rated scenarios (Fig. 10a and 10b), which have stronger wake effects (higher $C_t$). For the above-rated scenario (Fig. 10c), the directionally averaged turbulence intensity in FAST.Farm is very close to the freestream value for all turbines in the farm.

The comparison between FAST.Farm and the EFF is consistent across wind speed scenarios, with the EFF turbulence levels being a bit higher than the directionally averaged FAST.Farm levels. The differences in turbulence levels are largest for the innermost turbines, where wake effects are larger. For example, T4 has three neighbor turbines and has the closest match between both models. In contrast, T7 has eight neighbor turbines and has the farthest match. Another noteworthy remark is that, while the FAST.Farm directional average is consistently lower than the EFF predictions, the directional results from FAST.Farm are often higher than the highest effective turbulence levels by several percent. This is especially true at below- and near-rated wind speed for wind turbines in the wake when flow is aligned with a row in the farm. As expected, the results are also sensitive to the interturbine spacing. FAST.Farm clearly shows higher wake turbulence along vertical columns of wind turbines (where spacing is tighter at $5D$) than along horizontal rows (where spacing is $8D$).

We can obtain further insight into these results by considering the entire directional distribution of turbulence across the four quadrants (Fig. 11). This analysis reveals that while the effective turbulence is consistently higher than the FAST.Farm median



(a) Below-rated scenario

(b) Near-rated scenario

(c) Above-rated scenario

**Figure 10.** Hub-height turbulence intensity (%) at each wind turbine in the farm for each wind speed scenario. The FAST.Farm bars represent seed-averaged results for wind directions $0°$–$90°$ while the solid line is the directional average for wind directions $0°$–$360°$. The effective turbulence model (EFF) shaded region marks the range of results for the lowest ($m = 4$) and highest ($m = 10$) Wöhler exponent (the $m = 6$ values are in between). For reference, the average freestream turbulence ($I_{hub,obt}$) is also shown (dashed black line).



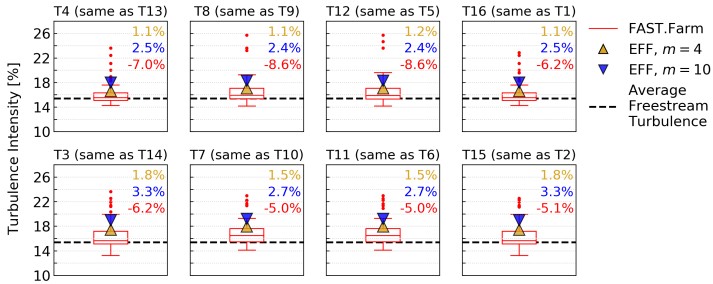

(a) Below-rated scenario

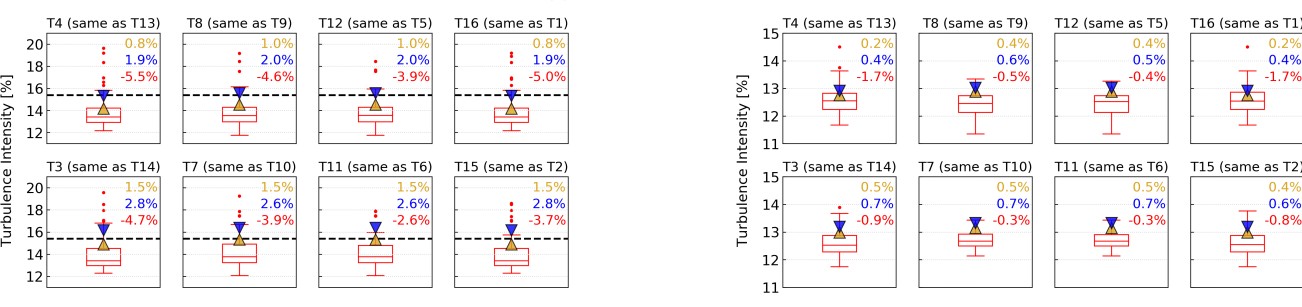

(b) Near-rated scenario

(c) Above-rated scenario

**Figure 11.** Hub-height turbulence intensity (%) at each wind turbine in the farm for each wind speed scenario. The FAST.Farm results are given in terms of boxplots (box spans inter-quartile range with an inner marking for the median) and outliers, with 72 seed-averaged values (one for each wind direction) included in each distribution. The effective turbulence model (EFF) results are given for the lowest ($m = 4$; yellow) and highest ($m = 10$; blue) Wöhler exponent (the $m = 6$ values are in between). For reference, the average freestream turbulence ($I_{hub,obt}$) is also shown (black dashed line). For brevity, only the top half of the wind farm is shown. Results are symmetric for the bottom half. The percent values given alongside each distribution quantify the difference between the EFF results and FAST.Farm statistics: the gold percent compares EFF $m = 4$ to the FAST.Farm median; blue compares EFF $m = 10$ to the FAST.Farm median; red compares EFF $m = 4$ to the FAST.Farm maximum.

values, it is always within the FAST.Farm distribution. Note that the FAST.Farm wake turbulence values go below the average freestream turbulence (the dashed black line in Fig. 11) for all wind turbines in the three scenarios. This happens because of lateral and vertical inhomogeneity in the undisturbed inflow. Recall that $I_{hub,obt}$ is simply a lateral average at hub height

through the farm-domain inflow fields. The actual value at a wind turbine location for a particular turbulent seed might be substantially lower as confirmed by the freestream turbulence distributions shown in Fig. 9. As a consequence, the differences seen between FAST.Farm and the EFF are due not only to wake effects but also to the level of fidelity in the turbulent inflow that drives the simulations: a single, precise number for the EFF and a dynamic, heterogeneous flow field for FAST.Farm. Finally, it is worth noting that the EFF only has four unique results for a set of 16 turbines (as was shown in Fig. 8); the other

12 turbines are mirrored due to the wind farm rectangular symmetry. This symmetry is broken in FAST.Farm by the lateral variations in turbulence along rows and columns, leading to eight unique results, as indicated in Fig. 11.




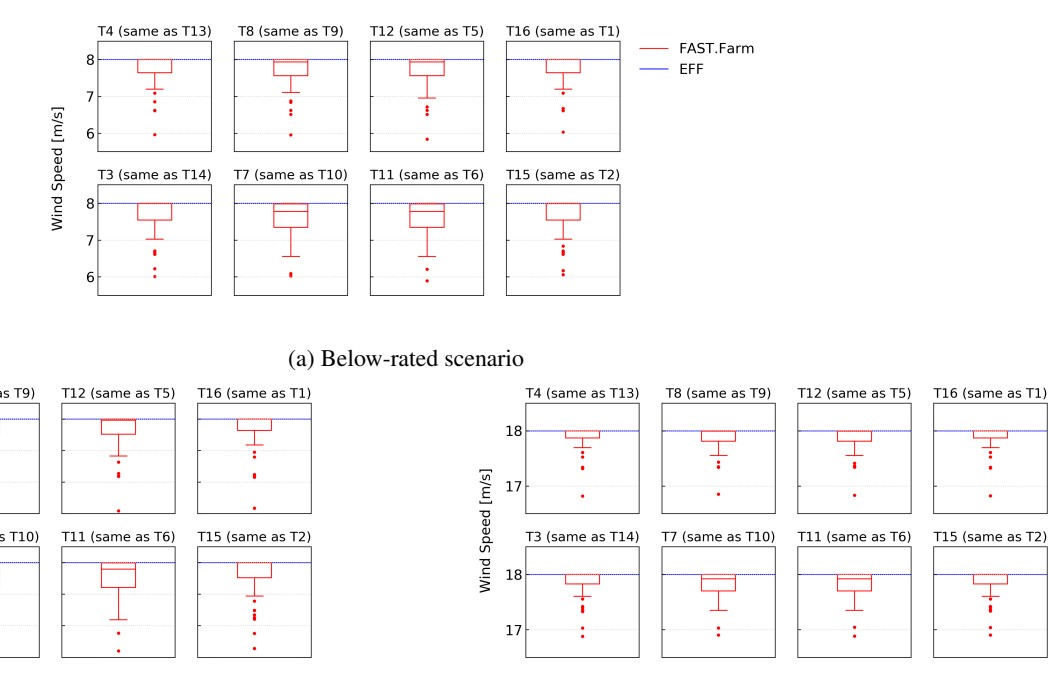

**Figure 12.** Mean wind speed (m s$^{-1}$) at each wind turbine in the farm for each wind speed scenario. The FAST.Farm results are given in terms of boxplots and outliers, with 72 seed-averaged values (one for each wind direction) included in each distribution. The effective turbulence model (EFF) results are given as the horizontal line, which is constant for all wind turbines and represents the undisturbed wind speed. For brevity, only the top half of the wind farm is shown. Results are symmetric for the bottom half.

As previously noted (Table 1), the EFF loads calculation is based on undisturbed wind speed values, which are 8 m s$^{-1}$, 12 m s$^{-1}$, and 18 m s$^{-1}$ for the three scenarios considered here. Conversely, the wind turbine loads computed by FAST.Farm are affected by the wake wind speed deficit. A quantitative assessment of the difference in wind speeds driving the loads in both methods is shown in Fig. 12. These distributions reveal that for certain wind directions the wind turbines in FAST.Farm experience mean wind speeds that are much lower than the freestream value: up to 2 m s$^{-1}$, 3 m s$^{-1}$, and 1 m s$^{-1}$ lower for the below, near, and above rated scenarios, respectively. In fact, the turbines at the center of the farm (T6, T7, T10, T11), which are waked for all wind directions, have a median seed-averaged wind speed that is slightly lower than the freestream values: 7.8 m s$^{-1}$, 11.8 m s$^{-1}$, and 17.9 m s$^{-1}$. The directional dependence of these results can be quantitatively seen in the one-quadrant roses of Fig. 13.

## 3.2 Wind turbine loads

The analysis performed up to now indicated overall higher mean turbulence with wakes for the EFF compared to FAST.Farm. Based on this result, it would be reasonable to expect that the EFF would consistently predict higher load variability than





(a) Below-rated scenario

(b) Near-rated scenario

(c) Above-rated scenario

**Figure 13.** Mean hub-height wind speed (m s$^{-1}$) at each wind turbine in the farm for each wind speed scenario. The FAST.Farm bars represent seed-averaged results for wind directions $0°$–$90°$ while the solid line is the directional average for wind directions $0°$–$360°$. The effective turbulence model (EFF) line is constant for all subplots due to the lack of wake wind speed deficits in the EFF modeling approach.





FAST.Farm. To assess this hypothesis, we consider the distributions of load standard deviations (Fig. 14), a quantity that we
use as a proxy for fatigue loads. We see that the median values of the load standard deviations are indeed generally higher
for EFF than FAST.Farm, as signaled by the mostly positive percent values above the distribution pairs. These percent values
represent the difference between the median standard deviation for FAST.Farm and the EFF. Differences higher than 5% are
shown in bold, revealing that the biggest discrepancies are for blade root out-of-plane bending moment, tower base fore-aft
moment, tower top side-side moment, and torque. Whenever the percent difference is negative (i.e., the FAST.Farm median
standard deviation is higher) the difference is low (i.e., $< 3\%$). Another obvious takeaway from Fig. 14 is the much larger
spread of the FAST.Farm distribution with long upper and lower tails. This is due to the wind direction effects, which are
explicitly present in the FAST.Farm results and only implicitly included in the EFF calculations as well as on the spatial
variation of turbulence in the FAST.Farm results.

The statistical analysis discussed above quantified the differences in load fluctuations between the two modeling approaches.
Next, we consider load power spectra to determine whether these differences in standard deviation are concentrated to a
particular frequency range. We focus on blade root and tower base moments and consider the below-rated and above-rated
scenarios (Figs. 15 and 16). We choose the below-rated scenario due to the importance of wakes at these lower wind speeds,
and the above-rated scenario because high winds are usually the main drivers of fatigue. We choose blade root and tower
base for having higher load variability magnitudes. At the blade root (Fig. 15), we see that the in-plane moment differences
are concentrated at the blade-passing frequency 1P ($\sim 0.14$ Hz below rated and $0.2$ Hz above rated). For the out-of-plane
bending moment, the large difference in distribution spread and median observed at below-rated wind speed in Fig. 14b is
concentrated primarily at the low-frequency end of the spectrum followed by 1P. Above rated wind speed, the differences in
out-of-plane moment are also largest at the low-frequency end, with small differences also observed for 1P, 2P, and 3P. At the
tower base (Fig. 16), we saw larger fore-aft moment standard deviation differences below rated than above rated (Fig. 14h). At
first glance, the spectra seem to indicate the opposite due to the large differences at low frequencies seen in Fig. 16b. However,
the differences are indeed larger below rated than above rated, as can be noted by the different order of magnitude in the
vertical axis between Fig. 16a and 16b. For both scenarios, the spectra reveal that the tower fore-aft moment differences are
concentrated at the low-frequency end of the spectrum with small differences near the first tower natural frequency ($\sim 0.32$
Hz) and 3P. The yaw moment spectral power differences are driven primarily by low frequencies followed by 3P.

In addition to load fluctuations, we now compare the mean load values between the two modeling approaches. Recall that the
vertical shear profile is the same for the two methods: a power law with a 0.2 exponent. Therefore, the main difference in the
mean flow fields is the wind speed magnitude: at a fixed height, it is constant for EFF and variable for FAST.Farm depending
on the wake presence and strength. As shown in Fig. 12, the mean wind speed value in FAST.Farm is close to the freestream
value, but the spread of the distribution is large with a long tail toward lower wind speeds. A similar result is seen for the
mean loads (Fig. 17): the median values (marked by circles above the bars in each subplot) are close for both methods, but
FAST.Farm 10-minute means are much more widely distributed than those of the EFF. The FAST.Farm distributions present a
large tail toward low values for all load quantities that are positive scalars. For the load signals that can take on positive and





**Figure 14.** Distribution of loads standard deviations ($\sigma$). Each subpanel gives the distributions for FAST.Farm (11,520 points: 16 wind turbines, 10 seeds, 72 wind directions) and the effective turbulence model (160 points: 16 wind turbines, 10 seeds). The percent values above each pair of distributions represents the difference in median standard deviation between both modeling approaches: $(\langle \sigma_{EFF} \rangle - \langle \sigma_{FASTFarm} \rangle)/\langle \sigma_{FASTFarm} \rangle \times 100\%$. The thick borders mark quantities for which power spectra are provided in Figs. 15 and 16.





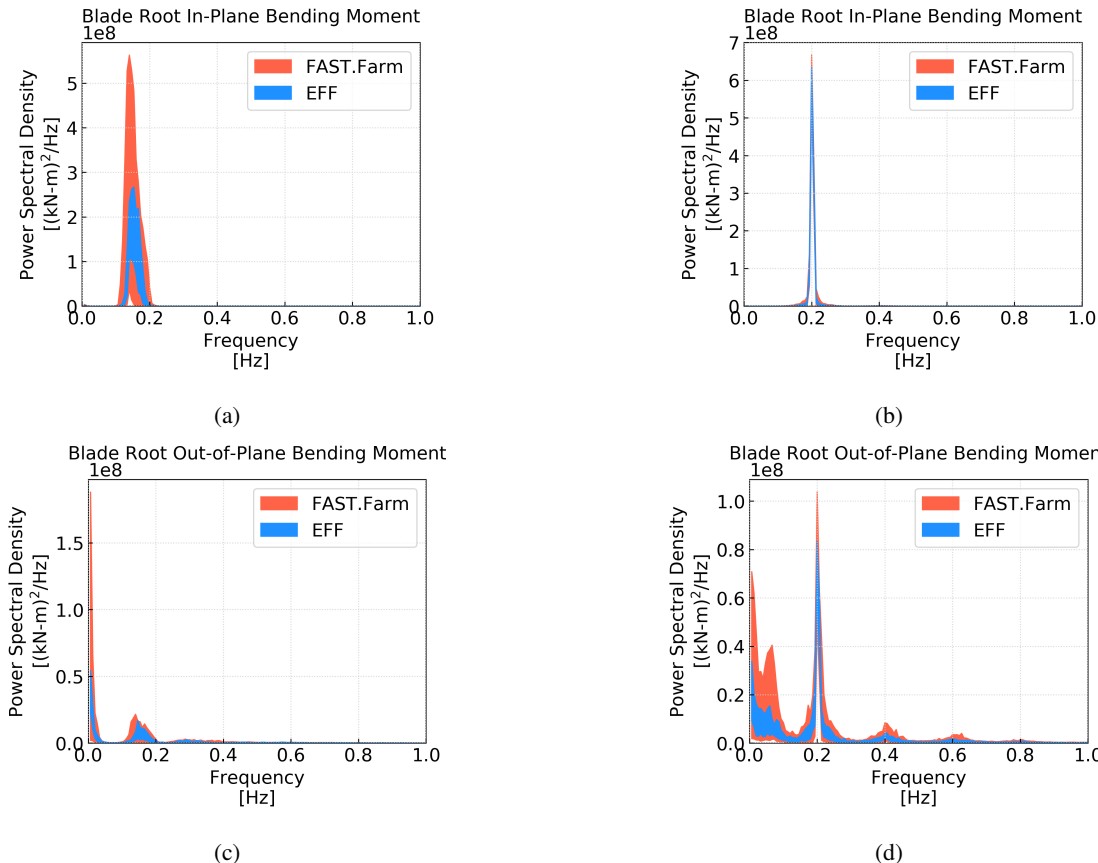

**Figure 15.** Power spectral density for the (a,b) blade root in-plane and (c,d) out-of-plane moments for the (a,c) below-rated and (b,d) above-rated wind speed scenarios. The spectra are first obtained for each 10-minute time series using a Welch's method and a Hanning window on 150-second segments. Next, range (minimum to maximum) of spectral power is plotted at each frequency.

negative values, the tails extend to both sides of the median. Overall, the differences between the two modeling approaches are less pronounced for the 10-minute mean loads than they were for the 10-minute standard deviations of loads.

**4 Summary and Conclusions**

The work presented here compares wind farm loads simulated with two different modeling approaches: the effective turbulence model and the dynamic wake meandering model as implemented in FAST.Farm. Both simulation strategies are provided as viable alternatives in the international standard for wind turbine design (International Electrotechnical Commission, 2019). The results from our comparison are summarized below in an effort to support intentional and informed decision making by

325 modeling experts working with wind turbine and wind farm design.



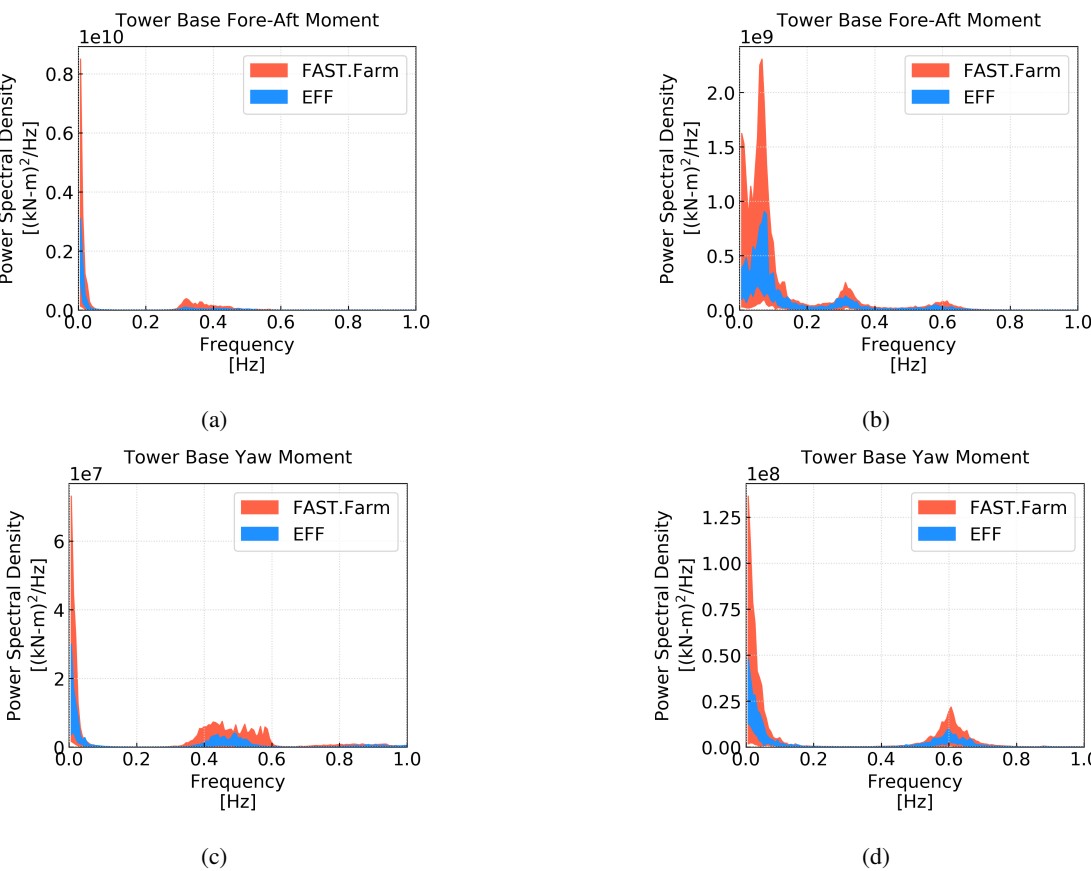

**Figure 16.** Power spectral density for the (a,b) tower base fore-aft and (c,d) yaw moments for the (a,c) below-rated and (b,d) above-rated wind speed scenarios. The spectra are first obtained for each 10-minute time series using a Welch's method and a Hanning window on 150-second segments. Next, range (minimum to maximum) of spectral power is plotted at each frequency.

For all cases considered, the EFF wind farm turbulence levels are consistently higher than the FAST.Farm medians computed across all wind directions ($0°–360°$ in $5°$ increments). However large, these EFF turbulence levels are still always within the directional distribution of FAST.Farm values. When comparing EFF to the FAST.Farm medians, turbulence intensity differences range between 0.2% and 2.7%, with FAST.Farm being lower. When comparing EFF to the FAST.Farm maximum

wind farm turbulence levels, differences reach up to 8.6%, with FAST.Farm being higher. The turbulence differences between both approaches are larger when the wake influence is strongest, which applies to wind turbines in the center of the farm and those in rows with tighter spacing. Wake-added turbulence, which is not accounted for in the FAST.Farm version used here, is not expected to have substantial effects on the results because the ambient turbulence levels considered here are already high (13.1%–16.5%). A previous study (Shaler and Jonkman, 2021) found that FAST.Farm matches large-eddy simulation results

well at high turbulence levels (i.e., greater than 10%) with differences seen at low turbulence levels (i.e., lower than 6%). On average, our results agree with previous studies that compared the EFF to measurements (Argyle et al., 2018; Reinwardt et al.,



**Figure 17.** Distribution of 10-minute means ($\mu$) for loads. Each subpanel gives the distributions for FAST.Farm (11,520 points: 16 wind turbines, 10 seeds, 72 wind directions) and the effective turbulence model (160 points: 16 wind turbines, 10 seeds).



2018) and DWM predictions (Reinwardt et al., 2018) and found EFF to overestimate turbulence levels. However, it is important to also note that the FAST.Farm wind farm turbulence is widely distributed and can, for certain wind directions and wind speeds, be substantially higher than that of the EFF. In terms of wind speeds, EFF compares well to the FAST.Farm directional

median but can differ by up to 3 m s$^{-1}$ when the FAST.Farm directional distribution extremes are considered. These large wind speed differences are due to the lack of wake deficits in the EFF formulation.

These flow field differences directly affect the load results. In terms of load means, the results are similar to the wind farm wind speed comparisons: EFF compares well to the FAST.Farm directional median but produces a narrower distribution of load means than FAST.Farm. The median of the EFF load standard deviations are overall higher than those of FAST.Farm,

with differences being largest for the blade root ($\sim 20\%$) and tower base ($\sim 17\%$) fore-aft moments below rated wind speed. The load spectra for these quantities show the largest differences at low frequencies, indicating that they are driven by the inflow. An example of a contrasting result is the in-plane blade root moment for which differences are concentrated in the first blade-passing harmonic. Our results indicate larger differences for blade root than for the tower, unlike the results of Reinwardt et al. (2018) and Schmidt et al. (2011b). In terms of the magnitude of these differences, it is hard to do a one-to-one comparison

of our results to previous studies due to the differences in simulation setups and analysis methods. However, we can say that our results broadly agree with Schmidt et al. (2011b) who found differences on the order of $10\%$ for blade flapwise moments and $20\%$ for tower base moments when comparing the EFF to a DWM implementation coupled to Bladed for the load calculations. The tower yaw moment, which was highlighted in a previous EFF-DWM comparison for being particularly sensitive to wind direction (Thomsen et al., 2007), remained within $\sim 7\%$ for the two modeling approaches.

Our analysis considered freestream turbulence, wind farm turbulence, load standard deviations, spectra, and means. It is clear from our results that substantial differences between both modeling approaches are already present at the very beginning of the simulation chain, before wakes are computed. As expected, these differences persist through to the loads results. The message is clear: on average, the EFF predicts higher turbulence and load variability, but grossly misses the wide distribution of turbulence and load levels that is found in wind farms not only due to wind direction variations with respect to the farm

layout, but also due to the spatial variability of turbulence itself even before wakes are taken into account.

The computational cost for running FAST.Farm instead of the EFF was 14 times higher for the inflow and 7 times higher for the wake and load simulations. Having said that, none of the simulations performed for this work required more than one computer node. Therefore, the added cost of running FAST.Farm is still arguably low and is not expected to hinder any future changes toward higher-fidelity design simulations.

Considering the largest differences we found (approximately 3 m s$^{-1}$ in 10-minute mean wind speed, 7% in 10-minute turbulence intensity, 20% in medians of 10-minute load standard deviations) and the arguably low increase in computational expense, we suggest that the wind industry seriously consider moving toward validated, higher-fidelity simulation tools such as FAST.Farm or other equivalent DWM approaches. By updating the wind turbine design paradigm, we have the potential to better optimize wind turbine designs or selections for conditions local to their installation site within a farm. In addition, this

change might prevent large unnecessary costs associated with overdesigning wind turbine components at sites in which design is driven by fatigue.





To complement the results presented here, future studies should focus on lifetime fatigue analysis and consider other wind turbine archetypes, realistic wind farm layouts, and nonuniform wind direction distributions. Inflow, turbine response, and wake measurements from a utility-scale wind farm would be valuable additions to the results we present. Finally, to support adoption

of higher-fidelity tools in design applications, we suggest that a standard procedure be developed through which DWM models can be applied in design and site suitability analyses.

## Appendix A: FAST.Farm Simulations Setup

The FAST.Farm parameters common to all simulations performed in this work are as follows:

- Type of inflow (`Mod_AmbWind`): 3 (i.e., separate synthetic farmwide and turbine-specific inflow domains)

– Time step for domains specific to wind turbines (`DT_High`): 0.1 s

- Number of planes used in wake calculations (`NumPlanes`): 175

- Number of radii in the radial finite-difference grid (`NumRadii`): 75

- Radial increment of radial finite-difference grid (`dr`): 5 m

- Vertical start of domains specific to wind turbines (`Z0_High`): 0.01 m

– Lateral and vertical start of farmwide inflow domain (`Y0_Low`, `Z0_Low`): $-2,325$ m, 0.01 m

The FAST.Farm parameters common to all wind farm orientations and turbulent seeds, but varying for the three inflow scenarios are given below. The three numbers provided for each quantity are given for the below-, near-, and above-rated scenarios, respectively.

- Time step and simulation duration:

– Simulation time (`TMax`): 1,100 s, 1,000 s, 900 s (only the last 600 s are used in the analysis, allowing for a spin-up of 500 s, 400 s, 300 s for each inflow scenario)

- Time step for wind farm domain (`DT`): 3 s, 2 s, 1.3 s

- Setup of the 16 domains specific to each wind turbine:

- Number of points in wind turbine domain (`NX_High`, `NY_High`, `NZ_High`): (33,31,34), (33,31,34), (29,31,34)

– Spatial resolution along $x$, $y$, and $z$ in wind turbine domain (`dX_High`, `dY_High`, `dZ_High`): (4.8,5,5) m, (4.8,5,5) m, (5.4,5,5) m

- Setup of the farmwide inflow domain:

- Number of points along $y$ and $z$ in wind farm domain (`NY_Low`, `NZ_Low`): (310,29), (232,22), (155,15).



– Spatial resolution along $x$, $y$, and $z$ in wind farm domain (dX_Low, dY_Low, dZ_Low): (12.325,15,15) m, (18.488,20,20) m, (27.720,30,30) m

## Appendix B: TurbSim Simulations Setup

When generating the farmwide inflows for FAST.Farm, the only varying TurbSim parameters across inflow scenarios are IECTurbc (set to $I_{mid,des}$, Table 2), URef (set to $V_{hub}$), NumGrid_Z and NumGrid_Y (Table A1). The TurbSim parameters common to all farmwide simulations are as follows:

– Time series length (AnalysisTime): 600 s

– Time step (TimeStep): 0.1 s

– Grid height (GridHeight): 420 m

– Grid width (GridWidth): 4,650 m

– Assumed hub height (HubHt): 210 m (while TurbSim refers to this as HubHt, we refer to it as the vertical center of the grid $z_{mid}$ to avoid confusion with the wind turbine hub height $z_{hub}$)

– Vertical wind shear (WindProfileType and PLExp): 0.2 power-law exponent

– Turbulence model (TurbModel): Kaimal spectrum (IECKAI)

– Spatial coherence model (SCMod1, SCMod2, SCMod3): IEC

– Spatial coherence parameters (IncDec1=IncDec2=IncDec3): (12.0, 0.0003527) is based on International Electrotechnical Commission (2019) recommendations with $a = 12$ and $b = 0.12/(8.1 * \Lambda_1)$ where $\Lambda_1 = 42$ m.

– Coherence exponent (CohExp): 0.0

**Table A1.** Farmwide inflow domain TurbSim parameters that varied across inflow scenarios: lateral grid size (GridWidth); number of points along vertical (NumGrid_Z) and lateral (NumGrid_Y) dimensions; spatial resolution of generated flow fields vertically $dz = $ GridHeight$/($NumGrid_Z$-1)$ and laterally $dy = $ GridWidth$/($NumGrid_Y$-1)$.

| URef=$V_{hub}$ | IECTurbc=$I_{mid,des}$ | NumGrid_Z | NumGrid_Y | $dz$ | $dy$ |
|---|---|---|---|---|---|
| (m s$^{-1}$) | (%) | (-) | (-) | (m) | (m) |
| 8.0 | 13.9 | 29 | 311 | 15 | 15 |
| 12.0 | 11.6 | 29 | 311 | 15 | 15 |
| 18.0 | 11.0 | 15 | 156 | 30 | 30 |



For the inflow generation specific to each wind turbine, the only varying TurbSim parameter across inflow scenarios is `URef` (set to $V_{hub}$). The TurbSim parameters common to all wind turbine simulations but different from the farmwide domain simulations are as follows:

- Grid height (`GridHeight`): 165 m

- Grid width (`GridWidth`): 150 m

- Number of points vertically (`NumGrid_Z`): 34

- Number of points horizontally (`NumGrid_Y`): 31

- Vertical resolution, `GridHeight`/(`NumGrid_Z-1`): 5 m

- Horizontal resolution, `GridWidth`/(`NumGrid_Y-1`): 5 m

- Turbulence model (`TurbModel`): Constrained turbulence (TIMESR)

The parameters `AnalysisTime`, `TimeStep`, `WindProfileType`, `PLExp`, `IncDec1`, `IncDec2`, `IncDec3`, `CohExp` match those used for the farm-domain simulations.

*Author contributions.* PD performed the simulations, conducted the analysis, and wrote the manuscript. KS wrote part of the code used to
430 perform the FAST.Farm simulations and revised the manuscript. JJ conceptualized the project, obtained funding for the work, participated in discussions about the methodology and results, and revised the manuscript.

*Competing interests.* The authors have declared no competing interests.

*Acknowledgements.* This work was authored in part by the National Renewable Energy Laboratory, operated by Alliance for Sustainable Energy, LLC, for the U.S. Department of Energy (DOE) under Contract No. DE-AC36-08GO28308. Funding provided by the U.S. Depart-
435 ment of Energy Office of Energy Efficiency and Renewable Energy Wind Energy Technologies Office. The views expressed in the article do not necessarily represent the views of the DOE or the U.S. Government. The U.S. Government retains and the publisher, by accepting the article for publication, acknowledges that the U.S. Government retains a nonexclusive, paid-up, irrevocable, worldwide license to publish or reproduce the published form of this work, or allow others to do so, for U.S. Government purposes.

The authors thank Ewan Machefaux of Vestas Wind Systems for suggesting the study.



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
