# Peer review of "Difference in load predictions obtained with effective turbulence vs. a dynamic wake meandering modeling approach"

_Wind Energy Science, 2023_

## Author Comment (AC2)

| | Reviewer Comment | Type of Response | Specifics of Response |
|---|---|---|---|
| 1 | L88: "CT is the wind turbine thrust coefficient" --> CT is the neighboring (?) wind turbine thrust coefficient | Revise text. | Revised text: "$C_T$ is the thrust coefficient of neighboring turbines. Here, $C_T$ is the same for the 16 wind turbines for a given wind scenario. In total, three individual wind and thrust values are used based on the scenarios defined in Sect. 2.3.1." |
| 2 | L102: "Note that wake-added turbulence is a forthcoming capability of FAST.Farm that was not available in the model version used here. That said, the ambient turbulence intensities simulated in the wind scenarios are high enough that the absence of wake-added turbulence would not likely impact the conclusions of this study (Shaler and Jonkman, 2021)."
To the best of my understanding, the EFF approach is based on the computation of an effective turbulence to consider the influence of the adjacent wind turbines on the target turbine, i.e. to consider the influence of the wake-added turbulence. However, regarding the FAST.Farm computations, you justify that the wake-added turbulence would not impact the conclusions because of the high turbulence wind scenarios. Since the wind scenarios are similar between EEF and FAST.Farm, I see a contradiction there. Could you comment on this? | Revise text. | Revised text: "Note that wake-added turbulence (the additional small-scale turbulence generated from the turbulent mixing in the wake) is a forthcoming capability…"

FAST.Farm does model the increased turbulence in the wake relative to the freestream, associated with the meandering wake deficits. The term "wake-added turbulence", despite common when discussing the dynamic wake meandering modeling framework, is indeed confusing. It doesn't refer to the total turbulence levels in a wind turbine wake relative to the freestream. Instead, it refers to small-scale added turbulence generated by the vortex breakdown and shear layer of the wake. The FAST.Farm manual describes it as "the additional small-scale turbulence generated from the turbulent mixing in the wake". |

| | | | |
|---|---|---|---|
| | L332: Same remark as before regarding the effect of the wake-added turbulence. | | |
| 3 | L210: I suggest adding a subsection regarding the structural loading computation inside the section "Methods" and refer to steps FF.3 and EFF.3 of Fig. 5. | Clarify to reviewer. | The load simulations are already explained in detail in "Simulation Approaches" (Section 2.1) which describes the effective turbulence methodology for standalone wind turbine load calculations (Subsection 2.1.1) and the FAST.Farm methodology for wind farm load calculations (Subsection 2.1.2). We believe it makes sense to introduce these two approaches first and then the details of the inflow, although in practice the inflow is generated first. So you are correct in that the order of the description doesn't match the order of the schematic in Figure 5. |
| 4 | L245: In Figure 10, some turbulence intensities obtained with FAST.Farm (directions 0° -90°) for non-waked turbines are much higher than the average freestream turbulence, e.g. T1 for 30° in Figure 10 (c). Considering T1, it is also the case to a lesser extent for other wind directions in Figure 10 (a) and 10 (b). Could you comment on this? | Clarify to reviewer. | This has to do with limitations of the turbulence simulation tool "TurbSim".

 First limitation: Given the height of our turbulence planes, we can only request a specific turbulence level at 215 m and not at 90 m. So we can't quite know what we're going to get at 90 m until we get it. When requesting a specific turbulence intensity at 215 m, we provide the target value but allow TurbSim to vary around that requested value. If you are familiar with TurbSim, that means the ScaleIEC option is set to 0 here. When ScaleIEC = 0, the turbulence intensity will have a Gaussian distribution about the target value. However, the value obtained can sometimes be more. That is what I am trying to show with Figure 9 – if you average all seeds, wind turbines, farm orientations you do end up with the requested value. But for specific turbines, |

| | | | orientations, seeds the freestream turbulence might indeed be larger than the target, as you point out. |
|---|---|---|---|
| 5 | L335: "On average, our results agree with previous studies that compared the EFF to measurements (Argyle et al., 2018; Reinwardt et al., 2018) and DWM predictions (Reinwardt et al., 2018) and found EFF to overestimate turbulence levels."

I found this sentence a bit in contradiction with one sentence of the introduction (L60): "The work that we present here was motivated by the small number of published studies on this topic and the lack of consistency among them." I therefore suggest slightly rephrasing to insist on the fact that your results agree with the specific works of Argyle et al., 2018 and Reinwardt et al.,2018 instead of using "agree with previous studies". | Revise text. | Revised text: "On average, our results agree with Argyle et al. (2018); Reinwardt et al. (2018) who compared the EFF to measurements and DWM…" |

All minor comments were addressed.

---

## Author Comment (AC3)

| | Reviewer Comment | Type of Response | Specifics of Response |
|---|---|---|---|
| 1 | Eq3: It would be helpful if C_T is expressed as a function of V_hub. | Clarify to reviewer. | Here, the $C_T$ was known so we are not using $7c/V_{hub}$ nor any other expressions to compute it from $V_{hub}$. |
| 2 | L 88: C_T is the thrust coefficient of neighbouring turbines. | Revise text. | Revised text: "$C_T$ is the thrust coefficient of neighboring turbines. Here, $C_T$ is the same for the 16 wind turbines for a given wind scenario. In total, three individual wind and thrust values are used based on the scenarios defined in Sect. 2.3.1." |
| 3 | L 103-105: The statement claiming that the absence of wake-added turbulence would not impact conclusions is quite bold. One of the main results of this paper is that the effective turbulence model results in higher turbulence levels than the dynamic wake meandering model, and at least some of this difference can be attributed to the missing wake-added turbulence. Please provide further insight on this matter. | Clarify to reviewer. | In prior work of the authors involving validation of FAST.Farm against large-eddy simulation results and physical measurements, we noticed that FAST.Farm accurately predicts turbulence levels in the wake when the ambient wind turbulence is high (above 10% TI), but not when the ambient turbulence is low (below 10% TI). The contribution of wake-added turbulence to the total turbulence level in the wake will be important in stable atmospheric boundary layer conditions when the ambient turbulence level is well below 10% TI. This work only considers cases with high ambient turbulence level, so we hypothesize that the lack of wake-added turbulence in the version of FAST.Farm used here is not a concern. An improved wake-added turbulence model is being added to FAST.Farm now, and once fully implemented and validated, can be applied to cases with any levels of turbulence to confirm our hypothesis. |
| 4 | L 137-139: It is not clear if the stated turbulence intensities are characteristic values. | Revise text. | Added a sentence: These turbulence values refer to the "characteristic" turbulence definition as per the international standard. |

| 5 | L 150: The end of bullet one requires editing since the text is in italics when it is not supposed to be. | Fixed. | |
|---|---|---|---|
| 6 | L 155-157: The choice of using 80% directly influences your results. Please explain how this choice impacts your conclusions, and if it is considered insignificant, provide an argument as to why. | Revise text. | A sentence was added: "We choose a high turbulence level to assess the difference between the two simulation methods when they are expected to differ the most in terms of fatigue estimates." |
| 7 | L 205-206: The numbers stated in the text differ slightly from those shown in Figure 8. | Fixed. | The numbers in Figure 8 were correct, the text was off by 0.01 m/s. |
| 8 | L 284-285: Load standard deviations are introduced rapidly. Please provide a more detailed explanation. | Revise text. | Added a sentence: "The standard deviations are computed over 10 minutes of load time series for each wind turbine, seed, and wind farm orientation." |
| 9 | L 290-293: Fatigue loads are heavily influenced by the highest load cycles (due to each load cycle being raised to the power of "m" when calculating its contribution to fatigue damage). It would be interesting to include a comparison of higher-order raw moments of the load standard deviation distribution as a supplement to comparing medians. | Clarify to reviewer. | We only looked at standard deviations as a proxy for fatigue in this work. Assessing fatigue more rigorously (e.g., rainflow counting, damage equivalent loads, Miner's sum, more load cases) could be done in future work. |
| 10 | L 342-345: Similar to the previous comment, please comment on the potential effect of narrow versus wide distributions. | Clarify to reviewer. | FAST.Farm predicts a wider variability in mean loads than ETM, with a trend toward lower mean loads for some load channels as a result of the lower mean wind speed for the waked turbines. A reduction in mean loads would have a net positive effect on fatigue because most materials can better withstand fatigue cycles at lower mean loads than |

| | | | they can when the mean loads are larger (based on the Goodman correction). |
| --- | --- | --- | --- |
| 11 | L 365-372: The industry is moving towards estimating fatigue loads by considering the entire ambient turbulence distribution rather than relying on the characteristic turbulence (i.e., integrate fatigue loads across the ambient turbulence distribution for each wind speed). This is intractable to do via aero-elastic simulation and therefore surrogate models are being developed. Such surrogate models are relatively easy to train for the effective turbulence as it does not require a lot of parameters – as opposed to DWM. It would strengthen the paper to briefly discuss this potential issue of integrating the DWM model into the current practice of wind farm design. | Clarify to reviewer. | We are not familiar with an industry trend toward considering the entire ambient turbulence distribution (the -1 standard does not consider probabilistic approaches to design). Regardless, this paper highlights the benefits that can be obtained by moving from simpler simulations (ETM) to more computationally expensive (but still tractable) simulations (FAST.Farm) in design. Certainly this move would be even more computationally expensive if the entire ambient turbulence distribution was used. That said, work is ongoing to develop surrogate models for loads that take into account wake effects (see Shaler, Kelsey, John Jasa, and Garrett E. Barter. "Efficient Loads Surrogates for Waked Turbines in an Array." Journal of Physics: Conference Series. Vol. 2265. No. 3. IOP Publishing, 2022 https://www.nrel.gov/docs/fy22osti/82524.pdf), which could aid the inclusion of wakes into a more probabilistic design approaches. |